# Simulation of Arc Discharge in an Argon/Methane Mixture, Taking into Account the Evaporation of Anode Material in Problems Related to the Synthesis of Functional Nanostructures

**DOI:** 10.3390/nano15010054

**Published:** 2024-12-31

**Authors:** Almaz Saifutdinov, Boris Timerkaev

**Affiliations:** Department of General Physics, Kazan National Research Technical University Named After A.N. Tupolev—KAI, Kazan 420111, Russia; btimerkaev@gmail.com

**Keywords:** arc discharge, methane, carbon, copper, unified model, extended fluid model

## Abstract

In this work, within the framework of a self-consistent model of arc discharge, a simulation of plasma parameters in a mixture of argon and methane was carried out, taking into account the evaporation of the electrode material in the case of a refractory and non-refractory cathode. It is shown that in the case of a refractory tungsten cathode, almost the same methane conversion rate is observed, leading to similar values in the density of the main methane conversion products (C, C_2_, H) at different values of the discharge current density. However, with an increase in the current density, the evaporation rate of copper atoms from the anode increases, and a jump in the *I*–*V* characteristic is observed, caused by a change in the plasma-forming ion. This is due to the lower ionization energy of copper atoms compared to argon atoms. In this mode, an increase in metal–carbon nanoparticles is expected. It is shown that, in the case of a cathode made of non-refractory copper, the discharge characteristics and the component composition of the plasma depend on the field enhancement factor near the cathode surface. It is demonstrated that increasing the field enhancement factor leads to more efficient thermal field emission, lowering the cathode’s surface temperature and the gas temperature in the discharge gap. This leads to the fact that, in the arc discharge mode with a cathode made of non-refractory copper, the dominant types of particles from which the synthesis of a nanostructure can begin are, in descending order, copper atoms (Cu), carbon clusters (C_2_), and carbon atoms (C).

## 1. Introduction

In recent decades, nanotechnology has become one of the most dynamically developing areas of science and technology, opening up new horizons for the development of materials with unique properties and functional characteristics. Among the various methods of synthesizing nanostructures, plasma synthesis stands out for its ability to create high-quality materials at the nanoscale using low-temperature plasma generated using gas discharges of various types. However, for controlled and repeatable synthesis, further improvement of methods and a deep understanding of the processes occurring in gas discharge plasma under the conditions of nanostructure synthesis are necessary [1,2,3,4,5,6,7,8,9,10,11]. Today, this task is one of the most important.

The key factors determining the process of nanostructure synthesis are temperature, pressure, buffer gas, the presence of vapors of structural elements, and the method of their preparation [1,2,3,5,6,7,8,9]. As is known, two main types of plasma can be generated using gas discharges: (1) highly nonequilibrium plasma, in which the temperatures of electrons and the heavy component differ by orders of magnitude, and (2) thermal plasma, in which the temperatures of electrons and the heavy component, if not equal, are close in value. This distinction allows us to differentiate between two main methods for synthesizing nanostructures in gas discharge plasma.

In the first method [12,13,14,15], based on the use of highly nonequilibrium plasma, impurities of various gasses are used as precursors, such as hydrocarbons, silanes, germanes, and others [12,13,14,15]. In this case, energetic electrons activate various channels of plasma-chemical processes in gas mixtures, leading to their decomposition and the production of atoms of carbon, silicon, germanium, and various types of radicals that self-organize and synthesize into different types of nanostructures.

In the second method, which relies on thermal plasma, usually generated by arc discharges [16,17,18,19,20,21], precursors are introduced into the discharge gap as a result of spraying and evaporation of working electrodes made from various materials, such as graphite, metals, and composites. Alternatively, precursors can be fed into the discharge area in the form of powder. The arc is characterized by a high temperature gradient (the maximum temperature in the arc core can reach 10,000 K and higher [22,23,24,25], while at the periphery, it can be several hundred degrees Kelvin). Therefore, most materials introduced into the arc discharge undergo phase transitions, evaporation, and atomization in the arc core, followed by rapid cooling of the products, which leads to condensation with a high level of supersaturation in the gas phase.

In recent years, there has been renewed interest in the use of arc discharge for the synthesis of various nanostructures, including functional core–shell particles [16,17,18,19,20,21]. This interest is due to the wide range of applications for such nanoparticles in various fields, including magnetic fluids [26], synthesis gas conversion [27], hydrogen evolution reactions [28], the synthesis of carbon nanotubes (CNTs) [29], and the creation of second-generation quantum devices, as well as biomedical applications such as drug delivery [30].

One of the key challenges in this area is the development of methods for controlling the size of metal particles during their synthesis. Experimental data show that the size of metal nanoparticles significantly affects their magnetic, optical, and catalytic properties, making this issue particularly relevant [31,32,33]. The formation of a carbon shell around a metal core is one approach to achieving the required control. For example, studies in chemical vapor deposition (CVD) have demonstrated that the growth of the metal core ceases after the carbon shell is formed [34,35].

Controlling the growth of metal particles is also critical in arc discharge-based carbon nanotube synthesis [5,8,10,36,37,38]. Studies indicate that catalysts larger than 5 nm facilitate the formation of multi-walled carbon nanotubes, while single-walled nanotubes require metal catalysts smaller than 3–5 nm [39,40,41]. In one study using DC arc discharge, Ni and Fe nanoparticles encapsulated in multiple layers of graphene were prepared and subsequently used as catalysts to grow multi-walled carbon nanotubes in a CVD reactor [29]. Another study demonstrated that carbon can penetrate the amorphous shell and reach the catalysts [35].

The synthesis of metal–carbon core–shell particles using methane arc discharge has been successfully implemented in several studies [42,43,44,45,46]. Some of these investigations explored the effect of methane on particle sizes. The authors compared the sizes of particles obtained by evaporating an iron anode at different partial pressures of CH_4_ impurity gas. It was found that the average particle size decreases with increasing partial pressure of methane [42]. Despite this observed dependence, its interpretation remains challenging. First, there was no direct comparison of particles created with and without methane impurity under the same conditions. Second, only the overall particle sizes were compared, which does not exclude the possibility that the changes could be caused by the thickness of the carbon shell. Finally, the experiments were conducted at different buffer gas pressures, which could affect the plasma characteristics and aerosol growth kinetics [47,48].

One study compared the sizes of carbon-encapsulated copper particles synthesized using He/H_2_ and He/CH_4_ gas mixtures. It was found that the presence of CH_4_ limits the aggregation of copper particles, resulting in a decrease in their size [49]. However, replacing a significant portion of the gas mixture with another gas may alter some arc parameters due to differences in gas properties, such as specific heat and thermal conductivity [49]. The variable particle sizes reported in the study complicate the interpretation of the obtained results.

A modified approach was proposed in one of the studies [50], in which carbon is formed in a thermal plasma by the decomposition of hydrocarbon gas (CH_4_) at the center of the arc, while an evaporating steel anode provides the sputtering of iron (Fe). The use of gaseous hydrocarbons instead of a consumable graphite electrode makes this method continuous and scalable for industrial production, which has already been successfully applied to the production of carbon black [51] and CNTs [52]. The authors demonstrated both experimentally and through modeling that the carbon coating indeed suppresses the growth of iron nanoparticles formed from iron atoms evaporated from the surface in an arc within an Ar/CH_4_ gas mixture. They showed that this effect is sufficient to obtain very small metal nanoparticles measuring only a few nanometers [50].

Despite extensive studies of arc discharges in the field of synthesizing metallic and functional nanostructures, the optimal plasma parameters for the formation of various types of nanostructures remain insufficiently explored. In this regard, it is extremely important to develop theoretical models of arc discharges that take into account complete nonequilibrium, a detailed set of elementary processes, and interrelated effects at the boundary of the gas discharge plasma and the electrode. In particular, it is necessary to consider the detailed kinetics of elementary processes in the gas discharge plasma, the heating of the electrodes, as well as the sputtering and evaporation of the electrode material into the plasma and the effect of these particles on the discharge characteristics. Conducting numerical calculations and simulations within the framework of such models will allow for predicting and determining the optimal parameters of arc discharges for the controlled synthesis of nanoparticles.

It should be noted that such models have been actively developed in recent years. In particular, models have been created that uniformly describe the processes occurring in the discharge gap and in the electrodes [53,54,55,56,57,58]. A detailed review of these and other models can be found in [59]. In [60,61,62], the processes of evaporation of carbon atoms into the discharge gap were studied, taking into account the assumption of local thermodynamic equilibrium. In [63,64,65], the evaporation of atomic and molecular carbon particles from graphite electrodes and plasma–chemical reactions involving them, as well as their influence on the discharge characteristics, were considered. In [66,67,68], simulations of an arc discharge were carried out, taking into account the evaporation of copper and tungsten particles into the discharge gap.

The presented work is a continuation of these studies and aims to investigate the parameters of arc discharge plasma in an argon/methane gas mixture, considering the evaporation of metal electrodes, using copper as an example, into the discharge gap, which is utilized to synthesize carbon–metal nanoparticles. In particular, a self-consistent model is proposed that describes an arc discharge in an argon/methane mixture with metal electrodes. Two fundamentally important cases are considered: arc discharges with a refractory tungsten cathode, and a non-refractory copper cathode.

## 2. Model Description

### 2.1. Model Equations and Boundary Conditions

To simulate the arc discharge, we used a model based on an extended fluid description of plasma, taking into account the processes occurring in the discharge gap and in the electrodes [65]. This model includes a system of continuity equations for various types of particles written in the diffusion-drift approximation, an equation for the balance of electron energy density, and an equation for the energy balance of the heavy component of the plasma. The Poisson equation is used to describe the self-consistent electric field in the plasma. In addition, as before, the heat conduction equations for the cathode and anode, as well as the continuity equations for the current density in electrodes, are incorporated into the model. The complete system of equations, as well as a detailed description of the boundary conditions, can be found in our previous work [65].

It is important to note that the article discusses two significant cases. In the first case, simulations were performed for an arc discharge with a refractory tungsten cathode and a copper anode, while in the second case, the focus is on an arc discharge with a non-refractory copper cathode and a copper anode. Since arc discharges are considered in the work, the dominant mechanisms of maintenance should be different from secondary electron emission. These mechanisms are caused by the high temperature of the cathode surface and the high value of the electric field at the cathode [69,70]. Moreover, in the case of a refractory and non-refractory cathode, they should be different. In the general case, the density of the emission current is determined as follows:
(1)
$$j_{TFE}\left(T_{c},E_{c},\varphi_{f}\right)=e\int_{-\infty }^{\infty }f_{d}(T_{c},W,\varphi_{f})D\left(E_{c},W\right)dW,$$

where, 
$T_{c}$
 is the temperature of the cathode surface, 
$E_{c}$
 is the electric field on the emitter surface, 
$\varphi_{f}$
 is the work function, which has the meaning of the electron energy component determined along the normal to the surface and counted from zero for a free electron outside the metal, and 
$f_{d}$
 is the energy distribution function of the cathode conduction electrons. For free electrons in a metal, the following is the Fermi–Dirac distribution [69]:
(2)
$$N(T_{c},\varphi_{f},E)=\frac{4\pi m_{e}kT_{c}}{h^{3}}\ln\left(1+\exp\left(-\frac{W+\varphi_{f}}{k_{B}T_{c}}\right)\right),$$


The probability of penetration is characterized by the transparency coefficient of the potential barrier, 
D
, which depends on the normal component of the electron energy to the emission boundary and the electric field at the cathode. For more details, see, for example, [69,70].

In the case where the main effect on electron emission is exerted by the high temperature of the cathode surface, it is assumed that the discharge is maintained by thermionic emission, and Equation (1) reduces to the Richardson–Dushman formula [69]. The current from the cathode surface is provided by the electrons of group IV in the diagram in Figure 1. In the case where the electric field near the cathode reaches high values, and the cathode is cold, then the main emission mechanism is field emission, which is described with sufficient accuracy by the Fowler–Nordheim formula [69]. In this case, the current is maintained due to the electrons of group I (see Figure 1).

However, in real discharges with both refractory and non-refractory electrodes, both effects—high temperature and high electric field strength on the cathode surface—can play a significant role. 

(1) In the case of a refractory cathode, which can withstand temperatures above the melting point of non-refractory materials, the emission current is mainly due to thermionic emission. However, in discharges at atmospheric pressure, the electric field strength on the cathode surface can also reach fairly high values. In addition, roughness and irregularities on the cathode surface can additionally enhance the electric field. In this case, group III electrons, due to thermal energy, jump over the barrier that has arisen as a result of the field action. This is the manifestation of the Schottky effect, which has a classical nature. As a result, the boundary condition on the cathode surface for the electron continuity equations must include the electron flow described by the Richardson–Dushman formula with the Schottky correction:
(3)
$$e\Gamma_{TE}\cdot n=A_{R}T_{c}^{2}\exp\left(-\frac{\left(\varphi -\sqrt{e^{3}E}\right)}{k_{B}T_{c}}\right).$$


(2) In the case of a non-refractory cathode, a high temperature of the cathode surface, of the order of the melting point, is reached fairly quickly in an arc discharge, in which case, intense evaporation of the cathode material into the discharge gap occurs. Since the ionization potential of metal atoms is much lower than the ionization potential of inert gasses, particularly argon, the intense ionization in the cathode region and the dominance of metal ions can lead to an increase in the electric field near the cathode surface. This fact leads to an increase and even dominance of the influence of the electric field, but the heating of the cathode surface will still play an important role. In this case, the process of maintaining the arc discharge must already be due to thermal field emission from the cathode surface. In particular, Ecker et al. in [71] showed that the thermal field emission current can be significantly enhanced by the field of ions coming from the plasma, without the need to increase the temperature or electric field on the cathode surface. Coulomb et al. [72] presented a numerical model of arc binding on a non-refractory cathode, taking into account the increase in pressure in the cathode region due to evaporation of the cathode material and the ion-enhanced thermal field emission of electrons. In addition, various protrusions can be present on the surface of a non-refractory cathode and thin films can be formed, which can also lead to an increase in the field at the cathode surface.

Thus, it is the electrons in group II (see Figure 1) that will ensure the maintenance of the arc discharge. These electrons have an increased probability of tunneling through the barrier, since they have thermal energy due to their high temperature [69]. The barrier for them is narrower and lower than for group I electrons. This process is called thermal field.

The thermal field emission current density cannot be expressed by simple formulas, and is instead determined by numerical methods [69]. Expressions for the thermal field emission current were first obtained by Gas and Mullin in the form of a cumbersome series; Murphy and Good [70] obtained integral formulas for the total current by summing the emission of electrons of all groups. Various methods for calculating integral formulas are presented in [73,74]. Various approximating formulas describing thermal field emission were obtained in [75]. The following expression turned out to be the most accurate for numerical calculations [74,75]:
(4)
$$e\Gamma_{TFE}\left(T_{c},\beta E_{c},\varphi \right)\cdot n=K\left(AT_{c}^{2}+B{\left(\beta E_{c}\right)}^{9/8}\right)\exp\left(-{\left(\frac{T_{c}^{2}}{C}+\frac{{\left(\beta E_{c}\right)}^{9/8}}{D}\right)}^{-0.5}\right),$$

where

$$K=1.45\frac{8100T_{c}^{3}+0.35{\left(\beta E_{c}\right)}^{2}}{9100T_{c}^{3}-5300\beta E_{c}T_{c}+0.75{\left(\beta E_{c}\right)}^{2}},A=\frac{4\pi q_{e}m_{e}k_{B}^{2}}{h^{3}},\;B=406\exp\left(-2.22\left(\varphi_{f}-4.5\right)\right),C=2.727\times 10^{9}\left(\varphi_{f}/4.5\right),D=4.252\times 10^{17}{\left(\varphi_{f}/4.5\right)}^{3},$$

where *K* is the correction factor, 
$T_{c}\;$
 is the cathode temperature, 
β 
 is field enhancement factor, 
$E_{c}$
 is the electric field of the cathode surface, 
A
 is the thermal field emission constant, and 
B,C,D
 are the thermal field emission correction factors associated with 
φ
. Expression (4) was used to determine the thermal field emission from the surface of a non-refractory cathode. Despite the fact that the formula is phenomenological, it allows us to describe the thermal field emission quite well [74,76]. Moreover, when describing the thermal field emission, the gain coefficient is used, which is actually a fitting parameter and is used in more rigorous mathematized approximations [71,72,73,74].

To describe processes involving evaporated atoms, the model was supplemented with a continuity equation for the density of copper atoms, as well as equations for the density of excited copper atoms and copper ions. The continuity equation for the density of neutral copper atoms,

(5)
$$\frac{\partial n_{Cu}}{\partial t}+\nabla \cdot \Gamma_{Cu}=S_{Cu},\;\Gamma_{Cu}=-\nabla \cdot \left(D_{Cu}\nabla n_{Cu}\right),$$

must take into account their evaporation from the electrode surface in the boundary condition. Within the framework of the equilibrium approximation, the presence of a Knudsen layer with vapor pressure 
$p_{\mathrm{sat}}\left(T\right)$
 was assumed. The mass transfer of the electrode material from the electrode to the gaseous phase was described using the Hertz–Knudsen–Langmuir equation [77],

(6)
$${\left.\Gamma_{Cu}\right|}_{c,a}=p_{\mathrm{sat}}\left(T_{c,a}\right)\sqrt{\frac{M_{Ci}}{2\pi k_{B}T_{c,a}}},$$

where 
$p_{\mathrm{sat}}\left(T\right)$
 is the equilibrium pressure of the metal vapor at the temperature of the electrode surface *T_c,a_*, which in turn is determined from the thermal conductivity equation for the electrode [65]. The vapor pressure of copper atoms at the surface of the electrode with a surface temperature *T_c,a_* was determined from the following relationship [67]:
(7)
$$p_{sat}=133.3[Pa]T_{c,a}^{1.27}10^{13.39-17,656/T_{c,a}},$$

Let us note once again that the evaporation of copper atoms was taken into account from the surface of the cathode if it was copper and from the surface of the anode.

As already noted, the electrode surface temperature was determined from the heat conductivity equations written as follows for the cathode and for the anode:
(8)
$$\rho_{c,a}c_{p\;c,a}\frac{\partial T_{c,a}}{\partial t}-\nabla \cdot \left(\Lambda_{c,a}\nabla T_{c,a}\right)=W_{joule\;c,a}.$$

Here, 
$\rho_{c,a}$
 is the mass density of the cathode or anode, and *c_p c,a_* is the specific heat capacity at constant pressure of the cathode or anode material. Λ*_c,a_* is the heat conductivity of the cathode or anode material, and *W_joule c,a_* is Joule heating by the current in the electrode. The boundary condition on the side with contact to the plasma is written as follows:
(9)
$$n\cdot \left(\Lambda_{c,a}\nabla T_{c,a}\right)=Q_{c,a}-Lm_{Cu}{\left.\Gamma_{Cu}\right|}_{c,a},$$

where *Q_c,a_* is denotes the heat flux from the plasma and is described in detail in our previous works [65], and the second term in right hand side describes the heat loss due to evaporation. The latent heat of evaporation *L* is expressed through the boiling *T*_b_ and critical *T*_c_ temperatures [67]. When the melting temperature *T_pl_* = 1356 K is reached, a phase transition occurs in the material of the copper anode or copper cathode, which is taken into account when solving Equation (8) using the apparent heat capacity method [65,66,68].

### 2.2. Elementary Processes in Argon Plasma

In a previous work [78], as well as in the studies conducted in [57,58], it was shown that a reduced set of elementary processes can be used to describe an arc discharge in argon. Therefore, when describing a discharge in argon, we used the elementary processes from [57,58]. In addition to electrons, the following types of particles were considered: 
$e^{-}$
, Ar^+^, Ar_2_^+^, Ar*, and Ar_2_*. The set of elementary processes involving argon is presented in Table A1 in Appendix A.

### 2.3. Kinetics of Elementary Processes Involving Methane CH_4_

To describe plasma–chemical processes involving methane and its conversion products, reaction sets from [13,65,79,80,81,82,83,84,85,86] were used. The following types of neutral particles, CH_4_, CH_3_, CH_2_, CH, C, C_2_, C_2_H, C_2_H_2_, C_2_H_3_, C_2_H_4_, C_2_H_5_, C_2_H_6_, H, H_2,_ ions CH_4_^+^, CH_3_^+^ CH_2_^+^, CH^+^, C^+^, C_2_^+^, H^+^, H_2_^+^, and H_3_^+^, and excited particles, CH_4_*, C*, and C_2_*, were taken into account. Preliminary simulations showed that under arc discharge conditions, the role of negative ions is small due to the high rates of detachment and recombination reactions; therefore, negative ions were not taken into account in the model. The cross-sections of processes involving electrons were taken from [80,84,85,86]. The complete set of reactions involving methane and its conversion products is presented in Table A2 and Table A3 in Appendix A.

### 2.4. Kinetics of Elementary Processes Involving Copper Cu

In the case of an arc discharge with a copper electrode, the evaporation of copper atoms into the discharge gap is expected. Electrons can participate in excitation and ionization reactions of copper atoms. Six excited effective states and one type of ion were taken into account, which are listed in Table A4 in Appendix A. The elementary processes taken into account in the model are presented in Table A5 in Appendix A. The set of elementary processes is taken from [66,87,88]. Note that all elastic and inelastic processes involving electrons are taken from the LXcat database [85,86].

## 3. Results and Discussion

Numerical simulations were performed for an arc discharge in an argon/methane gas mixture (with a ratio of 99.9% argon to 0.1% methane) under two scenarios. In the first scenario, the cathode was assumed to be refractory tungsten, while the anode was copper. The evaporation of copper atoms into the discharge gap was considered only from the anode. In the second scenario, both the cathode and anode were made of copper, and the evaporation of copper atoms into the discharge gap was assumed to occur from both electrodes. In both scenarios, a discharge gap *L_gap_* of 0.4 mm in length was examined, with the electrode length *L* was set at 20 mm. The calculations were performed using one-dimensional geometry. The schematic arrangement of the electrodes and the one-dimensional computational domain is shown in Figure 2. To determine the discharge current density, an electric circuit equation was solved, allowing for the discharge current value to be calculated. For this purpose, a voltage of 5 kV was set on the source and by varying the ballast resistance from 3 kOhm to several tens of Ohms, arc discharge modes were obtained in the current range from 1 to 100 Amperes. Since it was assumed that the current spot was uniform in radius and a one-dimensional approximation was used, this allowed for the current density to be calculated. When determining the discharge current density, it was assumed in the calculations that the radius of both electrodes was 2 mm.

Now, let us discuss the results of the numerical calculations. First, we will consider the results of the first scenario, where the cathode was refractory (tungsten) and the discharge was maintained by thermionic emission. Figure 3a shows the dependencies of voltage, cathode temperature, and anode temperature on the current density (with the recalculated current strength indicated on the upper scale). Figure 3b illustrates the dependencies of the densities of argon and the main products of methane conversion—atomic and molecular carbon, atomic hydrogen, and copper particles evaporated from the anode surface, along with their ions—on the current density.

It is evident that, in the range of current density values from 
$j=2.4\times 10^{5}$
 A/m^2^ to 
$j=4\times 10^{6}$
 A/m^2^ (or for the current strength range from *I* = 3 A to *I* = 50 A), a falling *I*–*V* characteristic is observed. In this case, the cathode surface temperature changes from 3000 K to 3700 K, and the anode surface temperature changes from 770 K to 1040 K. Starting from current density values from 
$j=4\times 10^{6}$
 A/m^2^ to 
$j=6.3\times 10^{6}$
 A/m^2^, the *I*–*V* characteristic begins to grow and then fall again. In this range, a sharper increase in the copper anode temperature is observed from 1040 K to 1430 K. It should be noted that at current densities of 
$j=5\times 10^{5}$
 and 
$j=5.5\times 10^{5}$
 A/m^2^, a jump in the *I*–*V* characteristics is observed, caused by a change in the plasma-forming ion, which is evident from Figure 3b. The discharge begins to “burn on copper”. This fact is due to the lower ionization energy of copper atoms compared to argon atoms.

It is evident from Figure 3b that the conversion rate in the current density range from 
$j=1.5\times 10^{5}$
 A/m^2^ to 
$j=4\times 10^{6}$
 A/m^2^ is approximately the same, and the distributions in the densities of carbon particles and hydrogen atoms do not change. At the same time, in the current density range of 
$j=4\times 10^{6}$
 A/m^2^ to 
$j=6.3\times 10^{6}$
 A/m^2^, the current–voltage characteristics exhibit decreasing dependencies that turn into uniform dependencies.

The density of evaporated copper atoms increases quite sharply in the range of current densities from 
$j=4.8\times 10^{5}$
 A/m^2^ to 
$j=5.5\times 10^{5}$
 A/m^2^, then a slower growth is observed in the range from 
$j=5.5\times 10^{5}$
 A/m^2^ to 
$j=5\times 10^{6}$
 A/m^2^ and saturation is reached, which is associated with the anode surface temperature reaching 1430 K, which exceeds the melting point of copper, but does not exceed the boiling point.

In this case, the density of copper atoms prevails over argon atoms. This solution is obtained as a result of the fact that the total pressure in the mixture is assumed to be constant.

Obviously, depending on the requirements for the synthesis of nanostructures, different values of the sizes of copper nanoparticles coated with carbon can be expected. For a more detailed answer to the question on the characteristic quantitative values of the sizes of copper nanoparticles, it is necessary to conduct a series of experimental studies.

Figure 4 and Figure 5 show the time dependences of the densities of neutral particles at the top and charged particles at the bottom for two values of discharge current: 10 A (current density 
$j=7.8\times 10^{5}$
 A/m^2^) on the left and 60 A (current density 
$j=4.8\times 10^{6}$
 A/m^2^) on the right, respectively.

It is evident that by the time 
$t=6\times 10^{-6}$
 s, a decrease in the argon density observed, which is associated with an increase in the gas temperature and the condition of constant gas mixture pressure. At the same time, in the time range from 
$t=6\times 10^{-6}$
 s to 
$t=4.4\times 10^{-2}$
 s, an intensive conversion of methane occurs. Atomic carbon and atomic hydrogen become the dominant impurity particles.

In the time interval from 
$t=5\times 10^{-3}$
 s to 
t=3.3
 s, an increase in molecular carbon (clusters) C_2_ is observed. It should also be noted that starting from the time 
t=0.1
 s, intensive evaporation of copper atoms from the anode surface begins. An increase in the density of copper atoms is observed up to a time of several seconds. Starting from the time 
t=3.3
 s, the densities reach saturation. In this case, the dominant impurity particles are atomic hydrogen and copper atoms. The second most important particles are atomic carbon and molecular carbon, respectively. For a current of 60 A, a qualitatively similar picture is observed. However, an increase in the density of copper atoms is observed over the entire considered time range up to 1000 s. Accordingly, it is copper atoms that are the dominant impurity atoms, followed by atomic hydrogen and almost equal densities of atomic and molecular carbon.

From the time dependences of charged particles (Figure 5), it is evident that up to the time of 1.6 s, the dominant type of ions was atomic argon, and from the time of 1.6 s, the dominant type of ions becomes the copper ion. At a current of 10 A, ions of argon and atomic and molecular carbon become the second most important. With an increase in the current density to 60 A, a similar picture is observed; however, the densities of ions of atomic and molecular carbon are higher in value compared to the argon ions.

Figure 6 shows the distributions of the main types of charged particles along the discharge gap for two values of current density, 
$j=7.8\times 10^{5}$
 A/m^2^ and 
$j=4.8\times 10^{6}$
 A/m^2^ (current of 10 A and 60 A, respectively). A narrow cathode layer and quasi-neutral plasma are observed. Along the entire length of the discharge gap, the dominant ions are copper ions for both values of discharge currents. At the same time, at a current of 10 A in the plasma region adjacent to the cathode layer, the second most important ions are molecular carbon ions and argon ions. However, when moving along the discharge gap towards the anode, a decrease in the density of argon ions is observed, and, further in the anode, the second most important ions are atomic and molecular carbon ions along the entire length of the discharge gap. However, when moving along the discharge gap towards the anode, a decrease in the density of argon ions is observed. From the center of the discharge gap to the anode, the second most important ions are atomic and molecular carbon ions.

When the discharge current increases to 60 A, the second most important ions are atomic and molecular carbon ions along the entire length of the discharge gap. It should be noted that their maximum density values decrease with an increase in current from 10 to 60 A, respectively, from 
$n=2.7\cdot 10^{20}$
 m^−3^ to 
$n=1.4\cdot 10^{18}$
 m^−3^.

At the next stage, simulations of the arc discharge in argon with an admixture of methane for a non-refractory copper cathode were carried out. It was assumed that the discharge is maintained due to thermal field emission. In this case, various field enhancement factors from 150 to 400 were considered. The dependences of the voltage on the discharge current density (and the current strength for an electrode radius of 2 mm) are shown in Figure 7a. They all have a decreasing shape. As can be seen, with an increase in field enhancement factor, the voltage drops across the discharge gap decrease.

In addition, Figure 6b shows the dependences of the cathode (solid lines) and anode (dashed lines) surface temperatures on the current density for various values of the field gain. It is evident that with the increase in the field enhancement factor, the temperature of the cathode surface decreases, which is caused by the increase in the value of electron emission and the sink in the heat flux density due to the emitted electrons.

Figure 8 shows the average values of the densities of the dominant particle types and their ions depending on the current density for different values of the field enhancement factor. It is evident that with an increase in the field enhancement factor, the current density value increases, at which the plasma-forming ion changes from argon to copper. In particular, for 
β=200
, copper ions become the dominant ion species at a current density of 
$j=1.75\times 10^{5}$
 A/m^2^, and for 
β=400
, copper ions become dominant at 
$j=2.2\times 10^{6}$
 A/m^2^.

This is again due to the fact that an increase in the field enhancement factor leads to an increase in thermal field emission, a decrease in the cathode surface temperature, and, as a consequence, a decrease in the evaporation of copper atoms into the discharge gap. It is noteworthy that for an arc discharge with a non-refractory anode (at all values of the field gain coefficient), the C_2_ clusters formed as a result of methane conversion predominate over the C carbon atoms. Since C_2_ clusters are precursors to the growth of diamond nanocrystals, the modes considered may be convenient for their synthesis, and copper atoms in this case may act as catalysts for the growth of nanodiamonds. On the other hand, under the conditions under consideration, the formation of copper carbides is possible; therefore, it is necessary to conduct systematic complex experimental studies.

Figure 9 shows the distributions in the densities of charged and main types of neutral particles, and Figure 10 shows the distributions in the temperature of the heavy component of plasma, the electron temperature and the electric field potential along the discharge gap at a current density of 
$j=7.8\times 10^{5}$
 A/m^2^ (corresponding to a current strength of *I* = 10 A) for two values of the field enhancement factor 
β=200
 and 
β=400
. It is evident that in the case of the field enhancement factor 
β=200
, copper atoms predominate after atomic argon, which indicates their intensive evaporation, with the dominant type of ions being atomic copper ions. The second most important ions are the ions of C_2_ clusters. It should be noted that, in this case, the potential drop is 50 V, the maximum gas temperature observed near the cathode region is 1860 K, and the cathode surface temperature is 1300 K.

With the field enhancement factor 
β=400
, the dominant type of ions are argon ions; this is due to the fairly effective thermal field emission from the cathode surface and the decrease in the temperature of the latter, which is 710 K. As a consequence, the evaporation of copper atoms from the cathode surface is greatly reduced. The maximum gas temperature in this case is 2200 K and is observed almost in the center of the discharge gap. It should be noted that since the role of thermal field emission of electrons increases significantly, the potential drop in the discharge gap, which is ~20 V, also decreases.

## 4. Conclusions

In this work, within the framework of a self-consistent model of arc discharge, the simulation of plasma parameters in a mixture of argon and methane is carried out, taking into account the evaporation of the electrode material in the case of a refractory and non-refractory cathode.

(1)It is shown that in the case of a refractory tungsten cathode, almost the same methane conversion rate is observed, leading to similar values of the density of the main methane conversion products (C, C_2_, H) at different values of the discharge current density. However, with an increase in the current density, the evaporation rate of copper atoms from the anode increases, and a jump in the *j*–*V* characteristic is observed, caused by a change in the plasma-forming ion. This is due to the lower ionization energy of copper atoms compared to argon atoms. In this mode, an increase in metal–carbon nanoparticles is expected.(2)It is shown that in the arc discharge mode with a refractory tungsten cathode, higher gas temperatures are observed in the discharge gap, while the dominant types of particles from which the nanostructure can begin to be synthesized in descending order are copper atoms Cu, carbon atoms C, and carbon clusters C_2_.(3)It is shown that in the case of a cathode made of non-refractory copper, the discharge characteristics and the component composition of the plasma depend on the field enhancement factor near the cathode surface. It is demonstrated that increasing the field enhancement factor leads to more efficient thermal field emission, lowering the cathode surface temperature and the gas temperature in the discharge gap. This leads to the fact that, in the arc discharge mode with a cathode made of non-refractory copper, the dominant types of particles from which the synthesis of a nanostructure can begin are, in descending order, copper atoms (Cu), carbon clusters (C_2_), and carbon atoms (C). In this mode, we should expect the growth of carbon nanoparticles, particularly nanodiamonds, with copper acting as a catalyst for their growth or copper carbides.

## Figures and Tables

**Figure 1 nanomaterials-15-00054-f001:**
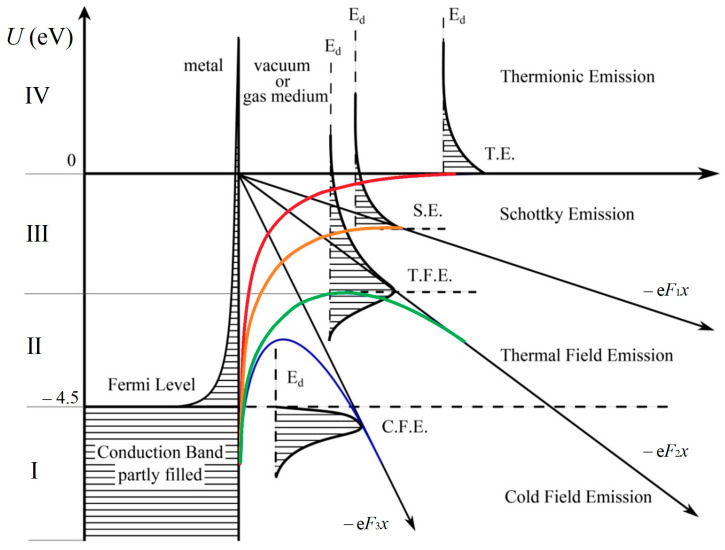
Electron spectrum in metal and different types of emissions.

**Figure 2 nanomaterials-15-00054-f002:**
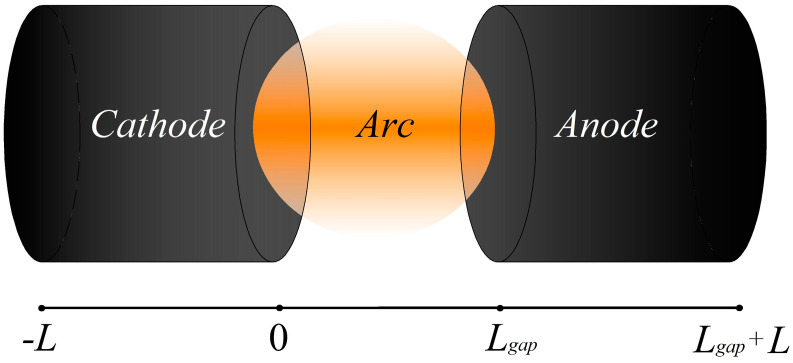
Schematic diagram of an arc discharge with the arrangement of electrodes (**top**) and the calculation area (**bottom**). The length of the electrodes is *L* and the length of the discharge gap is *L_gap_*.

**Figure 3 nanomaterials-15-00054-f003:**
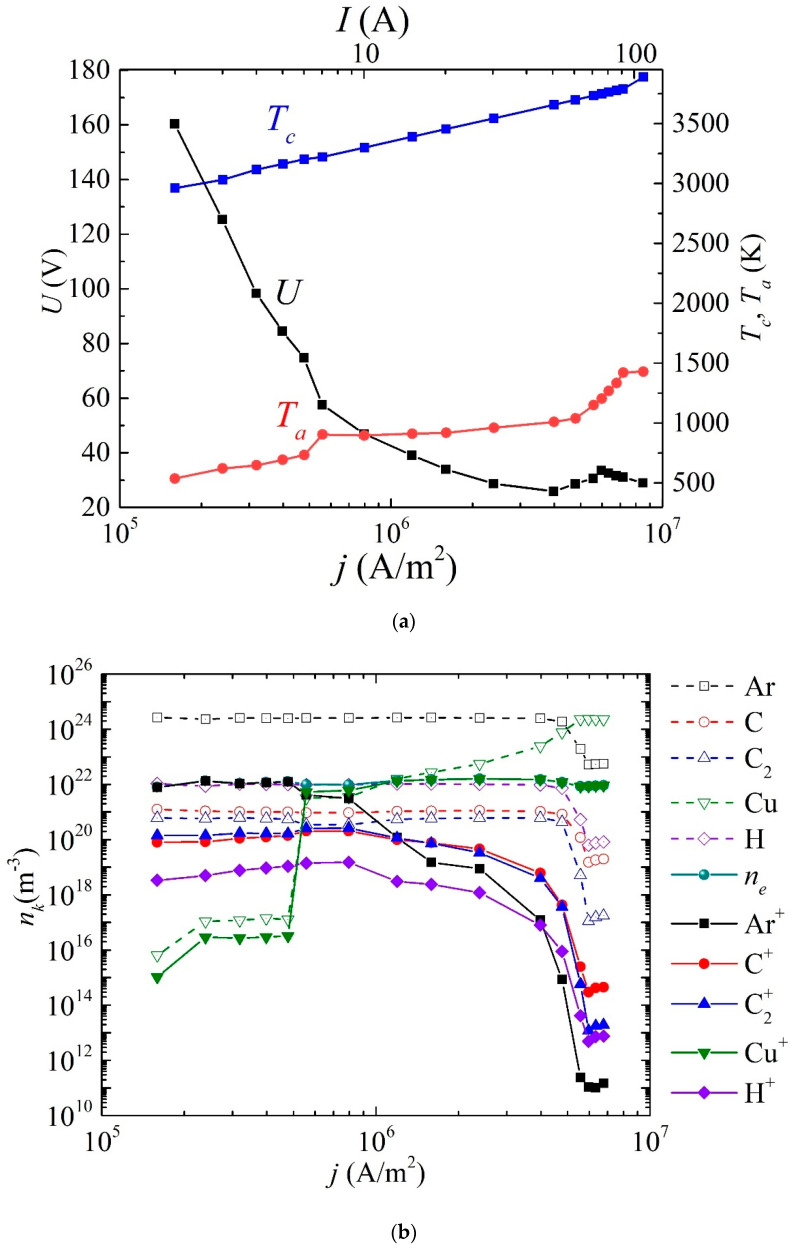
Dependence of (**a**) voltage, surface temperature of the cathode and anode, (**b**) averaged densities of dominant types of particles, and their ions over the discharge gap on current density.

**Figure 4 nanomaterials-15-00054-f004:**
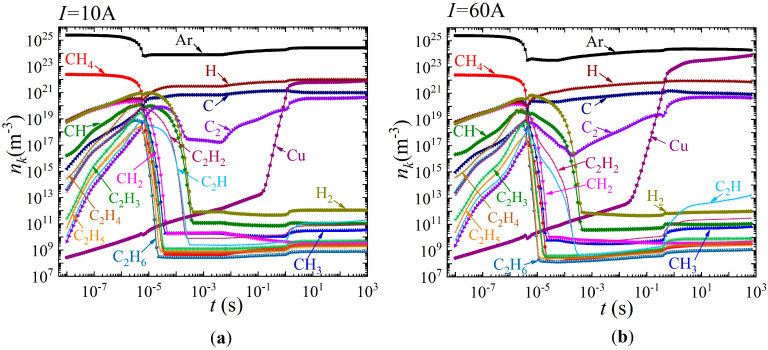
Dynamics of changes in the average densities of different types of neutral particles over the discharge gap at different values of discharge current, (**a**) *I* = 10 A and (**b**) *I* = 60 A.

**Figure 5 nanomaterials-15-00054-f005:**
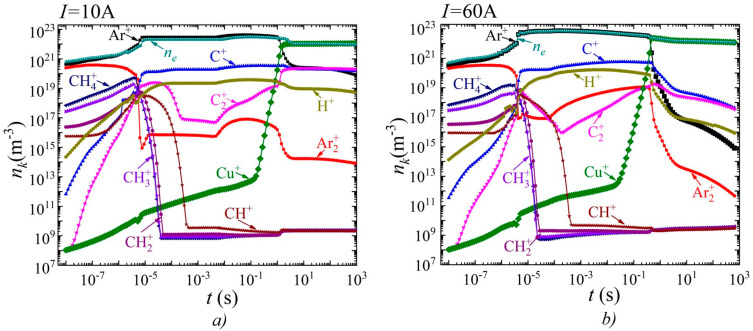
Dynamics of changes in the average densities of different types of charged particles over the discharge gap at different values of discharge current, (**a**) *I* = 10 A and (**b**) *I* = 60 A.

**Figure 6 nanomaterials-15-00054-f006:**
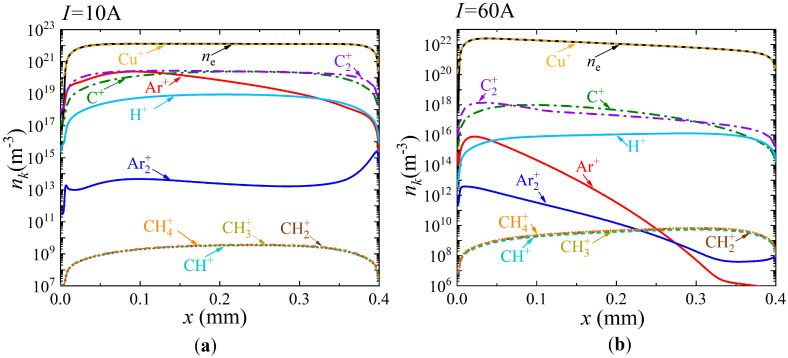
Distributions of charged particle densities along the discharge gap (the cathode is on the left and the anode is on the right) for two values of current density, (**a**) *j* = 7.8 × 10^5^ A/m^2^ (*I* = 10 A) and (**b**) *j* = 4.8 × 10^6^ A/m^2^ (*I* = 60A).

**Figure 7 nanomaterials-15-00054-f007:**
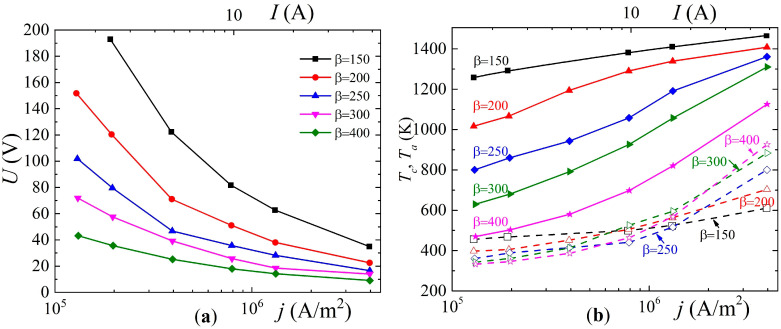
Dependences of (**a**) the voltage across the discharge gap and (**b**) the surface temperature of the cathode and anode on the current density for different values of the field enhancement factor near the cathode.

**Figure 8 nanomaterials-15-00054-f008:**
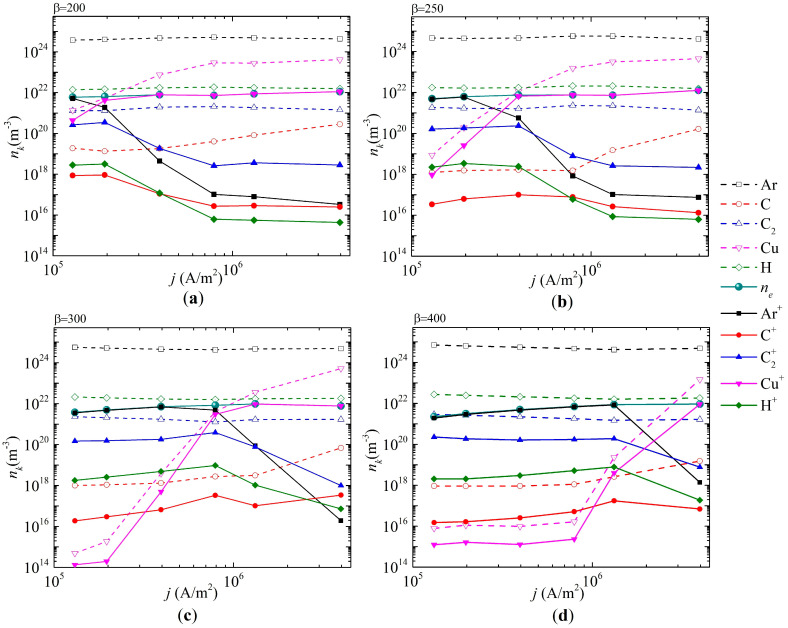
Dependences of the averaged densities of the dominant types of neutral and charged particles over the discharge gap as a function of the current density for different values of the field enhancement factor on the cathode surface: (**a**) 
β=200
, (**b)** 
β=250
, (**c**) 
β=300
, and (**d**) 
β=400
.

**Figure 9 nanomaterials-15-00054-f009:**
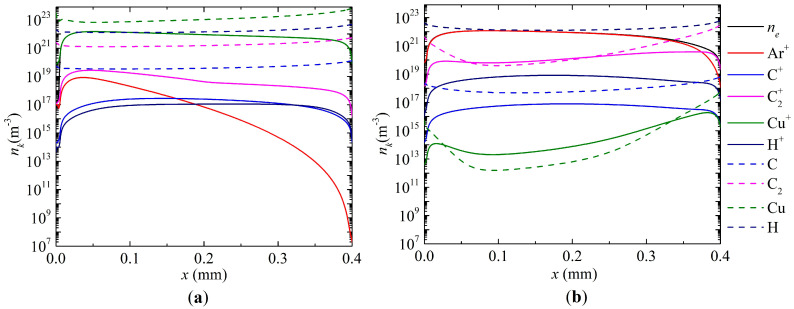
Distributions of densities of neutral and charged particles along the discharge gap (the cathode is on the left and the anode is on the right) at a current density of *j* = 7.8 × 10^5^ A/m^2^ for two values of the field enhancement factor, (**a**) 
β=200
 and (**b**) 
β=400
.

**Figure 10 nanomaterials-15-00054-f010:**
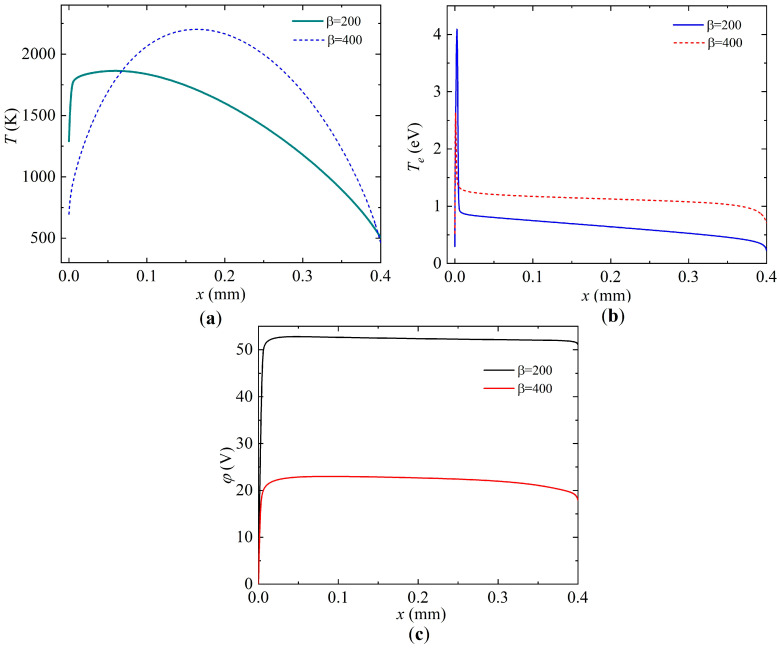
Distributions of (**a**) the temperature of heavy components of plasma, (**b**) the temperature of electrons and (**c**) the electric potential at a current density of *j* = 7.8 × 10^5^ A/m^2^ for two values of the field enhancement factor, 
β=200
 and 
β=400
.

## Data Availability

The datasets used and analyzed in the current study are available from the corresponding author on reasonable request.

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
