# Peer review of "Simulation of Arc Discharge in an Argon/Methane Mixture, Taking into Account the Evaporation of Anode Material in Problems Related to the Synthesis of Functional Nanostructures"

_nanomaterials, 2024, doi:10.3390/nano15010054_

Round 1
Reviewer 1 Report
Comments and Suggestions for Authors
I. Errors:
(1.1) There are errors in the descriptions of Figure 4 data, i.e., ions density distribution. At the current of 60A, the most dominant ions are Ar ions, but not copper ions. At current of 10 A, adjacent to the cathode layer, the second most important ions are molecular carbon ions and argon ions, not related to atomic carbon ions. Besides for these errors that need to addressed, the authors still need to stress that the maximum values of density given are related to the molecular carbon ions.
II. Drawbacks:
(2.1) In Figure 8, the curves of electron temperature and gas temperature near the cathode need to be enlarged, since the important value information is hidden in this superposed region.
(2.2) In Figures 2 and 3, the symbols for Ar, H, C, C2 and Cu, and their ion types, are better added along their respective curves. This is helpful for readers understanding their work.
III. Questions and comments
(3.1) Three types of current density are mentioned in the article, i.e., the continuity equation for current density in Page 3, the dependence of current density on temperature in Page 4 given by Eqs. (10) and (11), and the equation for an electric circuit used to determine the discharge current density in Page 12. Please, the authors address the difference and correlation among the three types of current density.
(3.2) The relation between the discharge current density and current intensity also needs to be addressed. In my opinion, the current density is spatially constant. Is that correct? Then it can be used to determine the current density, by means of the radius of electrode.
(3.3) Please briefly introduce the behind physics for the thermionic emission and thermofield emission. Since your article is mainly based on the two mechanisms, this introduction is certainly helpful for readers understanding your work, especially for the general researchers of low temperature plasma field.
(3.4) Please tell the difference and correlation between the electrode temperatures and gas temperature. What are their respective roles in the arc discharge? This is also helpful for understanding their work.
(3.5) With respect to the spatial resolution in Figures 4 and 7, please illustrate the positions of cathode and anode. Otherwise, it may be obscure.
(3.6) What is the external exciting parameter? In my opinion, the current density variation is triggered by an external parameter. Is that the voltage magnitude used in the boundary condition for Poisson equation? Please illustrate.
(3.7) Why aren’t there anions in the chemistry of argon and methane mixture? This is different to the argon and fluorocarbon mixed plasmas.
Author Response
We thank the distinguished reviewer.
We tried to respond to all comments and take them into account in the new version of the article. We believe that critical comments have significantly improved the quality of the article.
(1.1) There are errors in the descriptions of Figure 4 data, i.e., ions density distribution. At the current of 60 A, the most dominant ions are Ar ions, but not copper ions. At the current of 10 A, adjacent to the cathode layer, the second most important ions are molecular carbon ions and argon ions, not related to atomic carbon ions. Besides these errors that need to be addressed, the authors still need to stress that the maximum values ​​of density given are related to the molecular carbon ions.
Answer: Yes, thank you for the comment. Figure 4 presented incorrect data related to Figure 6. We have made a new graph and some corrections to the text of the article.
(2.1) In Figure 8, the curves of electron temperature and gas temperature near the cathode need to be enlarged, since the important value information is hidden in this superposed region.
(2.2) In Figures 2 and 3, the symbols for Ar, H, C, C2 and Cu, and their ion types, are better added along their respective curves. This is helpful for readers' understanding of their work.
Answer: Thank you for your comments (2.1) and (2.2). We have made corrections to the figures. In their current form, they look more clear to readers.
(3.1) Three types of current density are mentioned in the article, i.e., the continuity equation for current density in Page 3, the dependence of current density on temperature in Page 4 given by Eqs. (10) and (11), and the equation for an electric circuit used to determine the discharge current density in Page 12. Please, the authors address the difference and correlation among the three types of current density.
Answer: Thank you for your comment. We have made corrections to the text of the article.
(3.2) The relation between the discharge current density and current intensity also needs to be addressed. In my opinion, the current density is spatially constant. Is that correct? Then it can be used to determine the current density, by means of the radius of the electrode.
Answer: Thank you for your comment. Yes, you are right. In our case, the equation for the ballast circuit was additionally solved, V=V_{0}-I*R. The voltage on the source was set equal to 5 kV. The current strength and voltage drop across the discharge were determined by varying the ballast resistance. From the current strength, assuming a uniform current spot on the electrode, the total current density was determined.
(3.3) Please briefly introduce the behind physics for thermionic emission and thermofield emission. Since your article is mainly based on the two mechanisms, this introduction is certainly helpful for readers understanding your work, especially for the general researchers of low temperature plasma field.
Answer: Thank you for your comment. We have added to the text of the article to emphasize the difference between thermionic emission and thermofield emission.
(3.4) Please tell the difference and correlation between the electrode temperatures and gas temperature. What are their respective roles in the arc discharge? This is also helpful for understanding their work.
Answer: Thank you for your comment. In the work, for self-consistent modeling, the heating of the gas in the discharge gap was taken into account and the heat conduction equations for the cathode and the anode were solved separately. This was done to study the ignition of an arc discharge through a glow discharge and to study the dynamics of changes in the main products of methane conversion. That is, a glow discharge was ignited at a high current for short periods of time. Due to Joule heat generation, the gas in the discharge gap was heated. Then, due to the heat flow from the near-electrode regions, the electrodes were heated. Depending on the consideration of thermionic and thermoautoelectronic emission, the glow discharge transformed into an arc.
(3.5) With respect to the spatial resolution in Figures 4 and 7, please illustrate the positions of cathode and anode. Otherwise, it may be obscure.
Answer: Thank you for your comment. We have added an addition to the captions.
(3.6) What is the external exciting parameter? In my opinion, the current density variation is triggered by an external parameter. Is that the voltage magnitude used in the boundary condition for the Poisson equation? Please illustrate.
Answer: In our case, the equation for the ballast circuit was additionally solved, V=V_{0}-I*R. The voltage on the source was set equal to 5 kV. The current strength and voltage drop across the discharge were determined by varying the ballast resistance. The total current density was determined from the current strength under the assumption of a uniform current spot on the electrode. It should be noted that at first, we had a breakdown at short times and a glow discharge was ignited, while the current strength was high from units to several tenths of an ampere. Due to the high current, the Joule heating caused the heating of the gas and then the electrode surfaces. The discharge turned into an arc.
(3.7) Why aren’t there anions in the chemistry of argon and methane mixture? This is different from the argon and fluorocarbon mixed plasmas.
Answer: Thank you for your comment. This is a very correct question. In the initial calculations, we took into account negative ions. However, as preliminary numerical simulations showed, these anions disappear very quickly at short times, which is due to the high rate of electron detachment, as well as ion-ion recombination. In this regard, to speed up the calculations, we excluded them from the plasma chemical model.
In conclusion, we would like to once again thank the reviewer for his quality comments, which helped us improve the article.
Reviewer 2 Report
Comments and Suggestions for Authors
In this manuscript, according to the arc discharge self-consistent model, the plasma parameters in the mixture of Ar and CH4 were simulated from the perspectives of refractory electrode material (W) and non-refractory electrode material (Cu). It provides theoretical guidance for shape control and repeated production of nanoparticles prepared by the arc method. It is interesting and beneficial to actual operation.
(1) Self-consistent modeling is one of the important theories in this research, based on the analysis, what are the final phases in products by the arc discharge synthesis using the cathodes? Some of Tables 1-5, can put them as the appendixes at end of manuscript.
(2) Please provide the schematic diagram of arc-discharge setup, to show the electrodes and plasma state. The numbering of formulas in order needs well revision. In Eq. (24), the Hertz-Knudsen-Langmuir equation, how can distinguish the mass transfer formed by cathode solid-phase evaporation and by anode target evaporation? It should be further discussed.
(3) What is the difference between the Cu thermal field maintaining arc discharges and the W cathode surface thermal electron emission occurring discharges? Also needs an explanation.
(4) In Figs. 1-4, 6 and 7, the current density (J) of each element or charged particle, how to calculate or measure them? Also, the average density dynamics of different types for neutral and charged particles, the field enhancement factor β of the electrode, etc. these physical quantities should be accurately defined and make them clearer.
(5) The conclusion part needs reorganization, to be more reasonable and lucid.
Comments on the Quality of English LanguageThe English could be improved further more.
Author Response
We thank the distinguished reviewer.
We tried to respond to all comments and take them into account in the new version of the article. We believe that critical comments have significantly improved the quality of the article.
(1) Self-consistent modeling is one of the important theories in this study, based on the analysis, what are the final phases in products by the arc discharge synthesis using the cathodes? Some of Tables 1-5, can be put as appendixes at the end of the manuscript.
Answer: Thank you for your question. Our model is self-consistent and quite complex. At this stage, we were able to self-consistently take into account the formation of vapors of atomic and molecular carbon, as well as vapors of copper atoms. Of course, in the future it is necessary to take into account the formation of a solid phase in the discharge, which will allow a more accurate description of both the synthesis of nanostructures and the characteristics of the discharge. We agree with the comment about the tables. Therefore, we made an additional paragraph - an appendix.
(2) Please provide the schematic diagram of arc-discharge setup, to show the electrodes and plasma state. The numbering of formulas in order needs well revision. In Eq. (24), the Hertz-Knudsen-Langmuir equation, how can one distinguish the mass transfer formed by cathode solid-phase evaporation and by anode target evaporation? It should be further discussed.
Answer: Thank you for your comment. We took them into account and included them in the text of the article.
(3) What is the difference between the Cu thermal field maintaining arc discharges and the W cathode surface thermal electron emission occurring discharges? Also needs an explanation.
Answer: Thanks for the comment, we have added a more detailed description to the text of the article and the need to take into account thermionic or thermofield electron emission depending on the cathode material used in the problem.
(4) In Figs. 1-4, 6 and 7, the current density (J) of each element or charged particle, how to calculate or measure them? Also, the average density dynamics of different types for neutral and charged particles, the field enhancement factor β of the electrode, etc. these physical quantities should be accurately defined and make them clearer.
Answer: Thank you for the comments. Density in this case is the discharge current density. In addition, the dynamics of changes in neutral and charged particles were presented, and the beta coefficient included in the formula for thermal field emission was described in more detail.
(5) The conclusion part needs reorganization, to be more reasonable and lucid
Answer: Thank you for the comment. We have made corrections and structured the conclusion.
In conclusion, we would like to once again thank the reviewer for his quality comments, which helped us improve the article.
Round 2
Reviewer 2 Report
Comments and Suggestions for Authors
Authors have revised all issues raised by the referee, it becomes acceptable now.